# Why Do These Yeasts Smell So Good? Volatile Organic Compounds (VOCs) Produced by *Malassezia* Species in the Exponential and Stationary Growth Phases

**DOI:** 10.3390/molecules28062620

**Published:** 2023-03-14

**Authors:** Andrea Rios-Navarro, Mabel Gonzalez, Chiara Carazzone, Adriana Marcela Celis Ramírez

**Affiliations:** 1Grupo de Investigación Celular y Molecular de Microorganismos Patógenos (CeMoP), Universidad de los Andes, Cra 1 No. 18A-12, Bogotá 111711, Cundinamarca, Colombia; 2Laboratory of Advanced Analytical Techniques in Natural Products (LATNAP), Universidad de los Andes, Cra 1 No. 18A-12, Bogotá 111711, Cundinamarca, Colombia

**Keywords:** *Malassezia* volatile profile, volatile organic compounds, lipid-dependent yeast, *M. globosa*, *M. restricta*, *M. sympodialis*, growth phase, biological interaction

## Abstract

*Malassezia* synthesizes and releases volatile organic compounds (VOCs), small molecules that allow them to carry out interaction processes. These lipid-dependent yeasts belong to the human skin mycobiota and are related to dermatological diseases. However, knowledge about VOC production and its function is lacking. This study aimed to determine the volatile profiles of *Malassezia globosa*, *Malassezia restricta*, and *Malassezia sympodialis* in the exponential and stationary growth phases. The compounds were separated and characterized in each growth phase through headspace solid-phase microextraction (HS-SPME) and gas chromatography–mass spectrometry (GC–MS). We found a total of 54 compounds, 40 annotated. Most of the compounds identified belong to alcohols and polyols, fatty alcohols, alkanes, and unsaturated aliphatic hydrocarbons. Unsupervised and supervised statistical multivariate analyses demonstrated that the volatile profiles of *Malassezia* differed between species and growth phases, with *M. globosa* being the species with the highest quantity of VOCs. Some *Malassezia* volatiles, such as butan-1-ol, 2-methylbutan-1-ol, 3-methylbutan-1-ol, and 2-methylpropan-1-ol, associated with biological interactions were also detected. All three species show at least one unique compound, suggesting a unique metabolism. The ecological functions of the compounds detected in each species and growth phase remain to be studied. They could interact with other microorganisms or be an important clue in understanding the pathogenic role of these yeasts.

## 1. Introduction

*Malassezia* species are yeasts that normally inhabit mammalian skin, including human skin [1]. Currently, 21 species are suggested in the genus *Malassezia*, phylum *Basidiomycota* [2]. An important distinctive feature of all these species is that they are lipid-dependent due to the lack of genes encoding fatty acid synthase (FAS). This enzyme is required to synthesize de novo fatty acids, such as palmitic acid, a precursor of long-chain fatty acids needed for their growth [3,4]. Furthermore, *Malassezia* species can inhabit sebaceous areas, such as the scalp, back, face, and chest, where these yeasts intake lipids from their host [4,5]. To meet nutrient requirements, *Malassezia* is equipped with genes encoding lipases, phospholipases, and sphingomyelinases to metabolize fatty acids from the skin host and obtain nutrients to grow and proliferate [6]. A curious feature of this genus of yeasts is that the assimilation and presence of lipids differ between species, making them metabolically versatile [6].

*Malassezia* is considered an emergent and opportunistic yeast that takes advantage of changes in the host’s microenvironment, such as temperature, sweat, fatty acid level, and lifestyle, that could trigger that yeast to become pathogenic [1]. *Malassezia globosa*, *Malassezia restricta*, and *Malassezia sympodialis* are common species in human skin and are sometimes associated with dermatological diseases, such as pityriasis versicolor, seborrheic dermatitis/dandruff, atopic dermatitis, and psoriasis [5,7]. In addition, they are the principal species isolated from human skin, so it is appropriate to determine their secondary metabolites to start unraveling their strong interaction with the host [1]. Other species are related to fungemia, and recently, some authors have suggested an association of fungemia with Crohn’s disease [8] and Parkinson’s disease [9]. Furthermore, to try to understand why these yeasts transition from beneficial to opportunistic behavior, different approaches using in vitro or in vivo experimental models have been performed through several methodologies, such as omics tools, to elucidate the interaction between *Malassezia* and its host; these studies have been recently summarized [10].

The reasons why *Malassezia* becomes pathogenic remain unclear [3]. Small metabolites derived from its metabolism could be involved in commensal/pathogenic processes; this statement is proposed since all living organisms relate in some way to their environment, and *Malassezia* is no exception. These yeasts are a significant part of the skin microbiota, and they carry out both inter- and intraspecies interactions to obtain resources to grow and survive [11,12,13]. Microorganisms release volatile organic compounds (VOCs) to communicate with one another [14]. These small and odorous molecules are carbon-based compounds with low molecular weight, low boiling point, and high vapor pressure, and they are often in a gaseous state and spread in the environment quickly [13,15]. Chemical communication mediated by VOCs released by human microbiota is pivotal to understanding how this community of organisms cohabit and compete in the same habitat. The characteristic aromas from cheese, wine, yogurt, and even disgusting human odors are derived from microbial volatiles [16].

Several factors are involved in producing microbial VOCs, including substrates, interaction processes, type of ecosystem, temperature, and climate [17]. Additionally, species diversity, carbon energy sources, and pH are determining factors in VOC production [18]. Evidence suggests that microbial VOCs are generated from central and secondary metabolisms, such as glucose and fatty acid degradation [13,16]. Although detecting and separating volatile compounds is easily achieved through chemical techniques, such as gas chromatography (GC)-coupled mass spectrometry (MS) [19], the biogenic source of most of these compounds is not entirely understood [17].

Research on VOCs has been increasing because of their biotechnological/industrial applications, where volatiles are involved in the biocontrol of plant pathogens [20,21], clinical infections [22,23], biomarkers of diseases, and diagnosis [24,25,26,27]. Moreover, the roles of VOCs in biological and interactional processes have been described [28,29,30]. Currently, a VOC microbial database contains nearly 2000 volatile compounds registered as either bacteria or fungi. However, compounds from *Malassezia* are not available in this database [31]. Certainly, it is important to determine the volatile profile for *Malassezia* species and open the gate for further research on their implication in metabolism or biological interaction, as has been described for other microorganisms [32].

Currently, there are only two studies on VOCs produced by *Malassezia*. In 1979, the authors reported that these yeasts could produce VOCs on lipid-supplemented media; only 11 volatiles were identified [33]. Recently, 61 VOCs from *Malassezia furfur* were identified in growth media supplemented with oleic and palmitic acid, suggesting that fatty acid consumption determines VOC production in this species [12]. Nevertheless, the biological function of volatile compounds and their relationship with lipid metabolism and other metabolic processes remains unclear because there is no information concerning this topic [12,32].

This study aimed to identify the different VOCs produced by *M. globosa*, *M. restricta*, and *M. sympodialis* in exponential and stationary growth phases to clarify the metabolism of these yeasts. We found differences in the volatile profiles of these three species and identified at least one unique compound in each growth phase. These findings suggest metabolic changes in *Malassezia* associated with phylogeny and developmental phases. These novel results will allow species recognition based on a volatile profile and set a precedent for future studies about the role of VOCs from *Malassezia* yeasts in biological interactions with the skin microbiota or the host. Moreover, the volatile profiles help to understand why these yeasts could transition from being commensal to pathogenic.

## 2. Results

### 2.1. Growth Phases Differ between Malassezia Species

We evaluated growth across 12 sampling times (0–168 h) and counted colony-forming units (CFU) every day for 15 days. Based on growth curves for each species, we set sampling times for volatile organic compound (VOC) profiles accordingly in exponential and stationary growth phases. To determine each species’ exponential and stationary phases, a growth curve was generated in mDixon broth. The species *M. sympodialis* had a faster growth rate, reaching the exponential phase in 10 h and the stationary phase in 35 h. *M. globosa* and *M. restricta* took more time to reach each phase of development (Figure 1, Table 1), with *M. globosa* exhibiting slower growth.

### 2.2. The Volatile Profile Allows the Separation of Malassezia Species and Growth Phases

To identify volatile organic compounds (VOCs), we conducted headspace solid-phase microextraction and gas chromatography–mass spectrometry (HS-SPME/GC–MS) in the exponential and stationary growth phases of the three species. We found 54 compounds, 40 annotated (Table 2). Annotations of each volatile were performed, assigning retention times and comparing the experimental mass spectra with theoretical retention indices and the NIST Mass Spectral Library. Fourteen compounds released by *Malassezia* species were classified as unknown because it was impossible to make an accurate match of the experimental mass spectra with the NIST library or the compound was a possible isomer of other compounds, making it challenging to annotate them (Table 2). We isolated compounds detected in the mDixon media for further analysis.

Most of the volatile classes identified belong to alcohols and polyols, fatty alcohols, alkanes, and unsaturated aliphatic hydrocarbons (Table 2). Each species has unique compounds in its volatile profile (Figure 2). For instance, *M. globosa* produced six volatiles in both phases that were absent in the other species. Likewise, *M. restricta* and *M. sympodialis* released three differential compounds. In total, 29 compounds were shared by all species (Table 2, Figure 2). The total compounds characterized for each species with their corresponding identification number and the average percentage area, standard deviations, and the number of replicates in which each volatile was found are described in Table 2. The volatile compounds 3-propan-2-ylcyclohexene, octan-3-one, octan-3-ol, and 2-ethylhexan-1-ol were identified only for *M. globosa*, and 2,3-dimethylcyclopent-2-en-1-one was identified only for *M. restricta*. Finally, (E)-oct-3-ene and (3E)-octa-1,3-diene were identified only for *M. sympodialis*. *M. globosa*, in the stationary phase, released more unique compounds, such as hept-2-en-1-ol, octan-3-ol, and 2-ethylhexan-1-ol (Table 2).

Qualitative differences can be observed between all treatments where the abundance of compounds changes according to the species and the growth phase. The volatile profile emitted by the *Malassezia* species discriminated by growth phases, including the reference of the volatile organic compounds (VOCs) from the control, is shown in Figure 3. Moreover, it is appropriate to note that most peaks and some of the largest peaks belonging to the culture medium were used as a control and not attributed to the *Malassezia* species being examined.

Unsupervised and supervised statistical analyses were performed to determine if the volatile profile allows for the separation of the three species tested. First, the 3D-principal component analysis (PCA) explained 45% of the variance observed in the volatile profiles of *Malassezia* spp. For each species-growth phase, we plotted ellipses representing confidence intervals of 95% using a normal distribution to facilitate data interpretation. The first PCA dimension explained 23% of the total variance, which allowed for the separation of *M. sympodialis* in the exponential and stationary phases. Likewise, in this component, the differentiation of *M. globosa* and *M. restricta* in the exponential phase is evident (Figure 4). The other two dimensions of PCA explained 12% and 10% of the total variance, where it is possible to differentiate *M. globosa* and *M. restricta* in the stationary phase. PCA allowed for the partial discrimination and separation of the *Malassezia* species according to their volatile profile. In addition, chemical differentiation in the VOCs caused by the growth phase was observed. It was possible to note a high similarity between the three species in the stationary growth phase despite the high variation observed among replicates, especially for *M. restricta* in the stationary phase with a larger ellipse (Figure 4).

Second, supervised multivariate analyses were performed to confirm the chemical differentiation of the *Malassezia* species. Notably, 2D-projection to latent structures-discriminant analysis (PLS-DA) showed the discrimination of species, growth phase, and species/growth phase according to their VOC profiles. Figure 5a shows differences among species (R2X = 0.514, R2Y = 0.927, Q2 = 0.798), Figure 5b shows differences among growth phases (R2X = 0.319, R2Y = 0.954, Q2 = 0.877), and Figure 5c shows differences among species/growth phases (R2X = 0.58, R2Y = 0.896, Q2 = 0.727). In all cases, the Q2 metrics provided evidence of appropriate adjustment of the model.

In Figure 5a, discrimination in component 1 explains 19% of the variance and distinguishes *M. globosa* from the other two species; likewise, component 2 explains 11% of the variance and separates the volatile profile of *M. restricta* from those of the other two species. In Figure 5b, both phases were perfectly separated in two dimensions, explaining 22% and 10% of the total variance. In addition, the R2Y (0.954) and Q2 (0.877) values indicate that the model is valid, and the prediction accurately explains the importance of the growth phase in the volatile profile. Figure 5c confirms that species and growth phases can differentiate VOCs. The first component summarizes 23% of the variance, while the second explains 11%. In this plot, the first component showed that *M. restricta* and *M. sympodialis* were partially separated. In the second component, *M. globosa* in the exponential and stationary phases was discriminated from the other species.

To estimate which volatile compounds contribute to the differentiation pattern in the PLS-DA made by species/growth phase, variable importance in projection (VIP) scores was calculated. VIP scores indicate the importance of each variable in the projection used in a PLS-DA model. Factors with VIP scores greater than one are regarded as significant [35]. The VIP showed the 24 most important VOCs that better differentiate between species/growth phases (Figure 5d). The unknown seven compounds had higher scores and were released by *M. sympodialis* in the stationary phase. Unique compounds produced by a unique species, or even compounds produced in a specific growth phase (Table 2), usually presented higher VIP scores. For instance, non-3-en-1-ol (VIP approximately 1.6) was detected only in the stationary phase for all species, and (3E)-octa-1,3-diene (VIP approximately 1.2) was emitted only by *M. sympodialis* in both development phases (Table 2, Figure 5d). Likewise, octan-3-one (VIP approximately 1.3) is released only by *M. globosa* in both phases.

Additionally, *M. globosa,* in its stationary phase, also produced 2-ethylhexan-1-ol differentially, which is classified as a VIP of approximately 1.1. Moreover, *M. restricta* released the volatile 2,3-dimethylcyclopent-2-en-1-one in both the exponential and stationary phases annotated with a VIP above 1.2 (Figure 5d). Additionally, the VIP scores identified other compounds with high VIPs, such as carbon dioxide, butan-1-ol, 1-heptyl acetate, and 2-methylpropan-1-ol, which are present in all three species but vary in their relative abundance (Table 2, Figure 5d).

Statistical analyses to support PCA and PLS-DA were performed. Permutational multivariate analysis of variance (PERMANOVA) was carried out with 999 permutations using Euclidean distances to identify which variables (species, phase, or the combination species/growth phase) better explained the variance observed and which allowed for the separation of species according to the volatile profile. According to the PERMANOVA results, the total variance is better explained by the interaction of species/growth phase (R^2^ = 0.56, *p* = <0.001). Moreover, the other variables analyzed separately show significance in the variance observed, but the best adjustment was obtained by species/growth phase according to R^2^ values (Table 3).

PERMANOVA was performed with 999 permutations using Euclidean distances.

## 3. Discussion

This study is the first report on volatile organic compounds (VOCs) produced by *Malassezia globosa*, *Malassezia restricta*, and *Malassezia sympodialis* in the exponential and stationary growth phases. Headspace solid-phase microextraction (HS-SPME) and gas chromatography–mass spectrometry (GC–MS) identified 54 compounds released by all three species, and multivariate analysis provided evidence that both the species and growth phase influence the emission of VOCs. The growth stage of microorganisms determines their metabolic processes, which are influenced by environmental factors, such as nutrient availability, temperature, CO_2_ levels, and humidity. Additionally, the growth phase explains how microbes adapt to a niche [36,37].

These findings indicate that VOCs produced by the *Malassezia* species are differentially emitted at different developmental stages. They demonstrate the metabolic diversity of *Malassezia* spp. since the volatile profiles of the different species strongly differ. *M. globosa* produces more compounds than others; for example, octan-3-one is produced by this species in both phases but with a higher peak area in the exponential phase. Likewise, only in the stationary phase is the VOC octan-3-ol produced. These findings may be similar to what occurs in mushrooms, where it has been suggested that reduction of octan-3-one through the activity of an alcohol dehydrogenase produces octan-3-ol [38], so the yeast could be producing a compound at an early stage of growth and metabolizing it at another, demonstrating the importance of the growth phase in the production of VOCs. This alcoholic compound (octan-3-ol) was also identified in the fungus *Trametes versicolor* in a study to determine the electrophysiological responses of Coleoptera to VOCs [39], as well as in *Tuber* species, where this compound shows phytotoxic activity [40].

It is known that *M. globosa* exhibits a longer stationary phase than the other species evaluated in this study, so this feature could influence VOC production. Other compounds that are differentially expressed in *M. globosa* at the stationary phase are hept-2-en-1-ol and 2-ethylhexan-1-ol, both belonging to fatty alcohols, indicating that they are possibly produced during the degradation of the fatty acid pathway. These profiles also support the metabolic diversity of *Malassezia:* despite all the species being lipid-dependent, they assimilate the fatty acids from the host differentially [3]. It has been demonstrated that *Malassezia* presents differences in the assimilation profile of lipids, such as palmitic acid, by *M. globosa* and *M. sympodialis,* which show defects in its assimilation [6]. In addition, the lipidome of the species tested shows differences in the presence of some lipids, such as triacylglycerols, where the content in *M. globosa* is lower than that in *M. restricta* and *M. sympodialis*, and cholesterol, which is more abundant in *M. globosa* than in the other two species [41]. Despite no information about the implication of lipid metabolism in VOC production, there could be a relationship since the assimilation of lipids differs between species. According to the growth phase, it is possible to establish some similarities between species in the stationary phase; for instance, 2-phenylethanol and non-3-en-1-ol are released by all species only in the stationary phase. Likewise, 2-phenylethanol is a common volatile released by several microorganisms, such as *Saccharomyces cerevisiae, Klebsiella pneumoniae,* and other bacteria and fungi [11,42,43,44].

Moreover, this compound has been associated with interaction processes related to insect attraction and repulsion [11,44]. Curiously, *M. restricta* and *M. sympodialis* have more similar volatile profiles than *M. globosa*, according to the grouping given by multivariate analysis; however, the phylogenetic relationships show that *M. restricta* is close to *M. globosa* according to their genomes [6]. Since the growth times of *M. restricta* and *M. sympodialis* were the most similar, they likely influence the volatile profile, making these species more similar, sharing compounds such as 2-methylpropanal and butane-2,3-dione, which have also been identified in staphylococcal species [45]. Additionally, supervised and unsupervised multivariate analyses confirmed that the species and growth phase produced differences in the volatile profiles released by *M. globosa*, *M. restricta*, and *M. sympodialis.*

Comparison of the volatile profiles of the *Malassezia* species evaluated in this study with that previously reported for *M. furfur* [12] demonstrated similarities, and compounds such as carbon dioxide, butan-1-ol, pentan-2-one, 3-methylbutan-1-ol, 2-methylbutan-1-ol, hexan-1-ol, and heptyl acetate produced by *M. furfur* were identified in all the species in this study. Additionally, the influence of the growth phase on the volatile profile was determined; for instance, compounds such as 2,5-dimethylhexane are produced by *M. furfur* in the stationary phase when the medium is supplemented with palmitic acid, and this compound is identified in lower proportions in *M. globosa* and *M. sympodialis,* which present difficulty in the assimilation of this fatty acid [6], suggesting that lipid metabolism affects VOC production. A similar observation was made for the volatile 2-methylpropan-1-ol produced by *M. globosa*, *M. restricta*, and *M. sympodialis* in the stationary phase, with the highest peak corresponding to *M. furfur*, which produced the same compound in the same phase but in a lower proportion [12]. Other compounds that share all species, including *M. furfur* only in the stationary phase, are 3-methylpentan-2-one and non-3-en-1-ol. These results confirm the statement that the stage of development is strongly implicated in VOC production.

Volatiles are probably produced due to different pathways involved in processes related to a later stage of development. A previous study on volatiles in *Malassezia* also identified 2-phenylethanol and hexan-1-ol despite the differences in conditions [33]. These results also indicate differences between volatile profiles; for instance, gamma lactones identified in a study in 1979 [33] were not identified in our study, probably because the experiments used different culture media, such as Littman agar and Sabouraud-dextrose agar, and, as demonstrated by Gonzalez et al. 2019 [12], the type of culture medium used affects volatile production. Other differential conditions in that first study [33] were an incubation temperature of 37 °C in contrast to an incubation temperature of 33 °C in our study. Additionally, they supplemented the media with lecithin, oleic acid, and triolein, which probably affected the volatile profile considering that a recent study proved that volatile profiles are influenced by media supplementation [12]. Finally, the differences between the results of the study performed in 1979 with those of the study presented here performed in 2019 could be explained by the methodology employed since the sampling and identification through GC–MS are different between both studies [12,33]. These findings suggest that volatile profiles depend on the natural variability of the species, the stage of development, and the experimental conditions.

Curiously, some compounds frequently reported in other studies on microbial VOCs were also identified in this study, but only in the controls of culture media without yeast; for instance, broadly studied 1-octen-3-ol, which has been reported as a characteristic volatile of fungi [11,17,30,46,47,48], and the compounds dimethyl sulfide and dimethyl disulfide, which have been shown to be related to biological and microbial interactions [22,49,50], were identified in our control media. Some of these compounds were reported in a study on VOCs for *M. furfur* [12]. It is strange that compounds widely studied in bacteria and fungi, such as those mentioned above, were found in our controls, so an exhaustive review was conducted to reject any contamination in the control media or mistakes in the data analysis. We concluded that there was no contamination of microorganisms because there was no evidence of growth. In addition, the controls were performed at different times to guarantee randomization, and at all times, the controls were found to be sterile. Likewise, each chromatogram was analyzed independently, and the matches with the libraries were as rigorous as possible. A possible explanation, at least in the case of *M. furfur*, is that proper purification of the controls was not obtained in previous studies. Some volatiles have also been reported for *M. furfur*. Thus, our results indicate the possibility that in the work reported by our coauthors, some compounds were incorrectly reported to be present in the volatile profile of *M. furfur*. Appendix A summarizes the volatile compounds identified in our controls.

Considering that for *Malassezia* species, there is no information on the biogenic source of volatiles, we hypothesized that the metabolic differences found in these yeasts might be related to lipid metabolism because degradation of fatty acids is one of the pathways by which volatiles are produced [16]. For instance, intermediates of fatty acid metabolism are precursors of VOCs, such as alkenes and alcohols [16], emitting most *Malassezia* species in both growth phases tested. It is important to highlight that these intermediates include acetyl-CoA [51]. This important molecule participates in the modification of fatty acids by *Malassezia* [6], so the intake and metabolism of fatty acids are potential precursors of VOCs for these lipid-dependent yeasts. Figure 6 suggests different possible pathways by which VOCs are produced.

The complex metabolism of these yeasts could be involved in VOC production, as has been reported for other microorganisms, such as bacteria. Lactic acid bacteria produce VOCs, such as 3-methylbutan-1-ol and 2-methylbutan-1-ol, by enzymatic conversion of branched-chain amino acids [16]. These compounds are identified in all species of *Malassezia* in higher proportions. It is possible that during the production of these compounds by *Malassezia,* the oxygen condition decreases during static headspace sampling, forcing the yeast to use alternative pathways to metabolize glucose. Consequently, VOCs generated by anaerobic metabolism are emitted by all species. These compounds have also been reported in *Trichoderma* species and are important in plant development [52].

Butane-2,3-dione is produced in a similar manner, being emitted under conditions of low oxygen content in bacteria [16]. It is produced by *M. restricta* and *M. sympodialis* in the exponential phase. These findings are interesting since all three species produced CO_2_ as part of their aerobic glucose assimilation, so it is daring to think that when these yeasts consume oxygen in a closed atmosphere, they start metabolizing glucose through other pathways. Another likely remarkable pathway by which VOCs are produced includes the degradation of l-phenylalanine and l-tyrosine, whose oxidative processes result in the emission of 2-phenylethanol [51] produced by all three species of *Malassezia* only in the stationary phase. It would be worth performing studies to elucidate the production of this compound in this development step. In addition, 2-phenylethanol seems to be a common compound released by several microorganisms [11,42,43,44].

The identification of another VOC, 2,5-dimethylpyrazine, in all species mainly in the stationary phase (except *M. restricta*, which presents it in both phases) suggests that *Malassezia* likely metabolize L-tryptophan [53]. This volatile draws much attention because it has been reported that during infection, bioactive indoles, such as malassezin, indirubin, indololcarbazole, and formylindololcarbazole (ligands of the aryl hydrocarbon receptor (AhR)) produced by *Malassezia* from tryptophan, could be involved in melanogenesis. For instance, malassezin can cause the apoptosis of melanocytes [54]. This suggests that VOCs produced from tryptophan may play an important role in infection processes. To summarize, the results presented here show that VOCs are very important compounds produced in the main metabolic processes carried out by *Malassezia*. They could direct different interaction processes with the host, so it is necessary to pay more attention to these compounds.

Regarding the intent to determine which VOCs released by *Malassezia* have a possible ecological role, for other microorganisms, it has been reported that VOCs are important in interaction processes. For instance, 3-methylbut-3-en-1-ol and 2-methylpropan-1-ol are involved in biofilm formation in *Pseudomonas aeruginosa* and methicillin-susceptible *Staphylococcus aureus* (MSSA) [55]. Notably, 3-Methylbut-3-en-1-ol is emitted by *M. globosa* and *M. restricta* only in the stationary phase, and all species produce 2-methylpropan-1-ol in the same phase, even in *M. furfur* [12]. Moreover, 3-Methylbutan-1-ol is also implicated in the formation of bacterial biofilms [55], and it has been identified in biocontrol yeast [21] and in association with plants [56,57]; all *Malassezia* species produce it in two phases. It is possible that if these compounds are implicated in the interaction between *Malassezia* and the microbiota, they must be produced in a mature growth phase or even during growth. Other compounds related to the interaction between bacteria and fungi are sulfur and pyrazines emitted by *Pseudomonas aeruginosa*. These compounds influence the growth of *Aspergillus fumigatus* [22]. *Malassezia* releases similar compounds, such as 2,5-dimethylpyrazine. Compounds such as 2-methylbutan-1-ol produced by *Saccharomyces cerevisiae* or 2-methylpropan-1-ol produced by *Muscodor albus* have antifungal activity [40] and were also identified for the *Malassezia* species. Other volatiles identified for *Malassezia* in this study are 2-methylpropan-1-ol, hexan-1-ol, and 2-methylpropanal, and they are common compounds reported by several microorganisms and related to biological interactions [40,45,58,59]. Further research is required to elucidate the biological role of VOCs in biological interactions and the metabolism of *Malassezia*.

Finally, it is interesting to note that VOCs are responsible for the odor of fruits and food; for instance, octan-3-ol and octan-3-one generate a sweet odor [38], and microbial VOCs are probably responsible for the earthy smell and body odors of humans. Many mushrooms have distinctive odor profiles [17]. Moreover, the odors released by microbes influence the attraction of insects, functioning as pheromones; for example, the odors emitted by grape-born yeasts positively affect moth oviposition and help insects locate hosts [11]. If you smell *Malassezia* species cultures, you will note differences in the aromas between species. Some of them, such as *M. furfur*, present aromas similar to wine or fermented fruits. These odorous signatures from microbial volatiles have become a perspective in the food industry and biotechnological processes. For example, *Saccharomyces* and lactic acid bacteria are considered by winemakers as decisive factors influencing wine aroma and consumer preferences since their volatile compounds are responsible for the aroma and flavors of wine [60]. It will be worthwhile to study the differences in the scents of *Malassezia* and postulate their metabolic processes, such as fermentation, in the improvement of the flavors of products produced by the food industry.

## 4. Materials and Methods

### 4.1. Strains and Growth Conditions

The reference strains of *Malassezia globosa* CBS 7966, *M. restricta* CBS 7877, and *M. sympodialis* CBS 7222 were obtained from the Fungal Biodiversity Center (Westerdijk Institute, Utrecht, The Netherlands). The cultures of *Malassezia* species were maintained in modified Dixon agar (per liter: 36 g mycosel agar [BD, Franklin Lakes, NJ, USA], 20 g Ox bile [Oxoid, Basingstoke, UK], 36 g malt extract [Oxoid, Basingstoke, UK], 2 mL glycerol [Sigma-Aldrich, Saint Louis, MO, USA], 2 mL oleic acid [Carlo Erba, Val de Reuil, France], and 10 mL Tween 40 [Sigma-Aldrich, Saint Louis, MO, USA]) at 33 °C [61].

### 4.2. Growth Curve

A growth curve was generated to determine when both exponential and stationary phases occurred in each species. Strains were grown on mDixon agar at 33 °C for five days. Two loops of cells from each colony were suspended in 5 mL of sterile water + Tween 80/0.5% to adjust the inoculum at McFarland scale 2. Next, 3 mL of inoculum was added to 27 mL mDixon broth at 33 °C and 180 rpm for three days. To determine the species growth, 0.3 mL of the previous culture was added to 29.7 mL of fresh mDixon broth. The culture was incubated 12 different times for growth (0, 12, 24, 27, 30, 33, 36, 48, 60, 72, 96, and 168 h) at 33 °C and 180 rpm. Counting of colony-forming units (CFU) was performed by serial dilutions at each growth time; for this, 100 µL of the culture was added to 900 µL of sterile water + Tween 80/0.5%. Subsequently, 100 µL of each dilution was inoculated on mDixon agar and incubated at 33 °C. Growth was monitored daily for 15 days. The experiments were performed in triplicate [12].

### 4.3. VOC Production

According to the growth curve (Table 1), the exponential and stationary phases were determined for each species: *M. globosa* 30/90 h, *M. restricta* 30/55 h, and *M. sympodialis* 10/15 h for each phase. The volatiles were obtained at these times. A 125 mL PTFE Erlenmeyer flask was used for yeast growth, with incubation under the same stirring and temperature conditions described for the growth of this yeast. Lids were used to seal the Erlenmeyer hermetically so that volatiles did not escape. The uninoculated mDixon broth was used as a control. They were analyzed in 5 experimental replicates [12].

### 4.4. VOC Sampling and GC–MS Analysis

To extract the extensive range of polarities of volatile organic compounds (VOCs), compounds were collected for 20 min by exposure to the headspace of each treatment (maintained at 33 °C, 90 rpm) on a solid-phase microextraction (SPME) fiber with a 50 µm/30 µm divinylbenzene/carboxen/polydimethylsiloxane (DVB/CAR/PDMS, gray) coating (SUPELCO, Bellefonte, PA, USA). Subsequently, VOC analysis was performed by desorption in the GC HP 6890 Series using an Agilent BP-5 capillary GC column (30 m × 0.25 mm × 0.25 µm, SGE, Austin, TX, USA) coupled to an Agilent Mass Selective detector 5973 (Agilent Technologies, Palo Alto, CA, USA) with helium as the carrier gas at a flow rate of 1.3 mL/min. Volatiles absorbed by the SPME fiber were desorbed by heating to 250 °C using splitless injection. The column temperature gradient was programmed to start at 40 °C for 0.5 min and then heated at 6 °C/min from 40 to 60 °C. Then, the temperature was elevated to 150 °C at 3 °C/min, finally to 250 °C at 10 °C/min, and held for 6 min (total run time = 40 min). The gas chromatography–mass spectrometry (GC–MS) filament source and the quadrupole temperature were set at 230 and 150 °C, respectively. Ions were generated using electron ionization (EI) (70 eV) and acquired in full scan mode at 2 scans/s over m/z 30–300. All samples, including linear alkanes, were run under the same chromatographic conditions. Linear alkanes of the series C8–C20 were used to determine retention indices (RI) and later for the tentative assignment of compounds [12].

### 4.5. Data Analysis

Data acquisition was performed from the volatile profiles of five biological replicates for each treatment using MSD ChemStation D.02.00.275 (Agilent Technologies), and automatic integration was performed using a threshold of 14 units. The reported abundance values are relative abundances of compounds, obtained by integrating the signal in their chromatographic peaks. Putative annotation of the aligned compounds was conducted by searching spectra against NIST MS 2.0 with the NIST 14 database and comparing the experimental retention index (RI) with the RI of compounds reported in the literature. Only those peaks absent in any of the five replicates of each uninoculated control were considered to be volatiles released by the *Malassezia* species. In addition, compounds found in only one replicate across all treatments were discarded. This exclusion criterion was selected as a compromise to guarantee some level of reproducibility while also allowing for some biological variability often found in experiments involving live biological systems [62]. Finally, compounds that did not provide an accurate match with the library were considered unknown.

After automatic integration, the obtained results were rigorously reviewed, and peak areas were used to construct the original matrix in which tentatively annotated VOCs were reported as columns (variables) and estimated peak areas were reported as rows (observations). As grouping variables, we included the three tested *Malassezia* species (*M. globosa*, *M. restricta*, *and M. sympodialis*), both growth phases (exponential and stationary), and the combination species/phase. Data were processed using a Hellinger transformation suitable for reducing the horseshoe effect in matrices with many zeros [63]. Then, Pareto scaling was applied, which allows for scaling of the data obtained from the chromatographic analysis, reducing the relative importance of large values but keeping the data structure partially intact [64,65].

Statistical analyses were performed to establish relationships between tested variables and to correlate how important each one is to the outcome where dependencies exist for dimension reduction of variables [66]. Permutational multivariate analysis of variance (PERMANOVA) was applied to a dataset as a nonparametric method to conduct multivariate ANOVA and test for differences between variables [67]. Afterward, multivariate analysis was conducted using two different approaches (unsupervised and supervised). The unsupervised method is principal component analysis (PCA), which allows for the visualization of complex data while reducing its multidimensionality [66]. Supervised multivariate analyses, such as discriminatory analysis, where the measured variables (VOCs) are considered the predictor variables and those that define object classes are treated as the response variable (grouping variables), were performed [66]. Projection to latent structures-discriminant analysis (PLS-DA) is based on the principle of partial least-squares (PLS) regression, which aims to predict response variables (Y) from a broad set of predictor variables (X) [68]. The output of PLS provides measures of the model fit (R^2^), model predictive power (Q^2^), and model accuracy based on a cross-validation procedure [66].

Moreover, other statistical features could be evaluated, such as the variable importance in the projection (VIP) score of a variable, which is calculated as a weighted sum of the squared correlations between the PLS-DA components and the original variable. The weights correspond to the percentage variation explained by the PLS-DA component in the model. The number of variables in the sum depends on the number of PLS-DA components found to be significant in distinguishing the classes. Factors with VIP scores greater than one are regarded as significant [35]. These statistical analyses were performed to analyze the volatile profile differences comparing species, growth phases, and the combination of species/growth phases. The statistical analyses were carried out using R studio software (http://www.rstudio.org/) (accessed on 15 April 2022) with the following packages: HybridMTest, vegan, MetaboAnalyze, psych, ropls, ggplot2, VennDiagram, FactomineR, Rcmdr, Plot 3D, and rgl. All of them are available in the repository of the R studio software (https://CRAN.R-project.org) (accessed on 15 April 2022).

## 5. Conclusions

In this study, we report for the first time a volatile profile of three species of *Malassezia* in two stages of growth, and the similarities and differences in their profiles. *Malassezia* species produced differential VOCs in the exponential and stationary growth phases, and these compounds were discriminated in the dimensional space given by multivariate analysis. *M. globosa* produced a higher quantity of volatiles than the other species tested, demonstrating the metabolic versatility in the metabolism of these yeasts, especially in regards to lipid metabolism, as some of the volatiles identified here are probably the result of the degradation of fatty acids. Additionally, all species share compounds that are associated with interaction processes. Further research is necessary to clarify the role of VOCs produced by the *Malassezia* genus since these compounds could be involved in interaction processes and could be important for understanding the pathogenic roles of these yeasts.

## Figures and Tables

**Figure 1 molecules-28-02620-f001:**
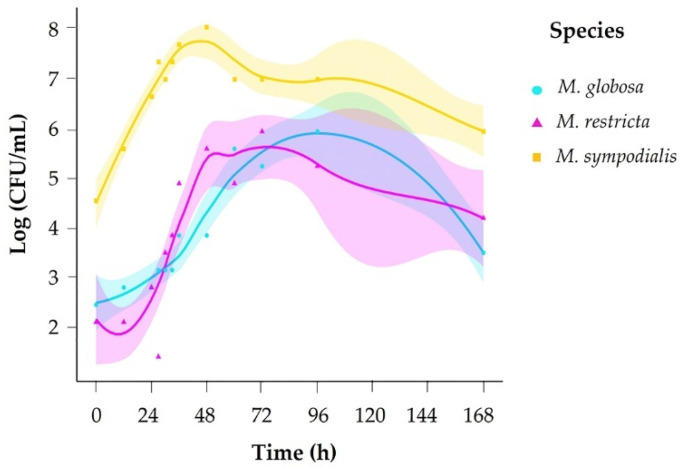
Growth curve for *Malassezia* species in mDixon broth.

**Figure 2 molecules-28-02620-f002:**
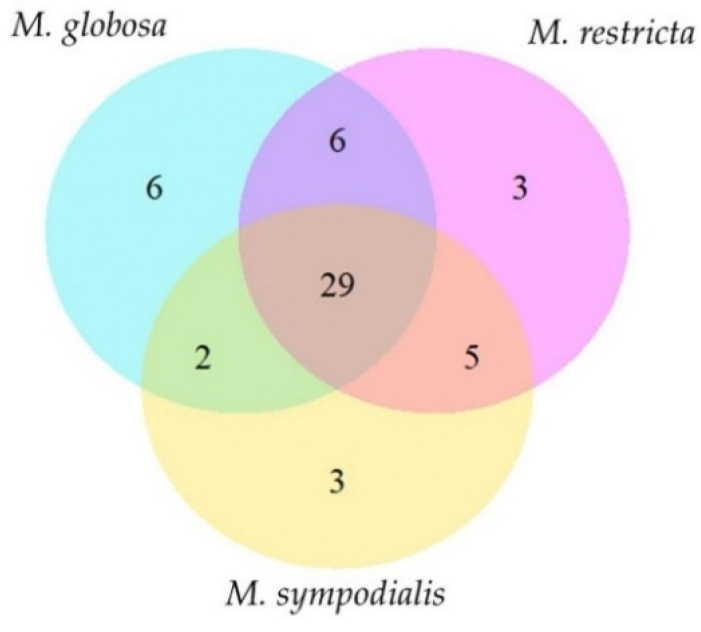
Venn diagram of the volatile profile of *Malassezia* spp. The diagram indicates unique and shared compounds between *Malassezia* species. Fifty-four volatile organic compounds were identified for all species.

**Figure 3 molecules-28-02620-f003:**
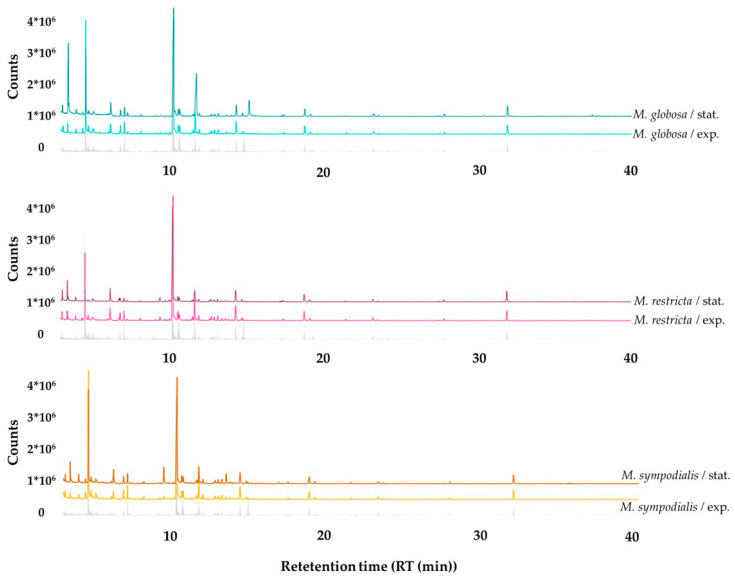
Gas chromatographic profiles of volatile organic compounds (VOCs) detected in the headspace of *Malassezia* spp. in the exponential and stationary growth phases. The gray line corresponds to the volatile profile of the control, which is mDixon media without yeasts.

**Figure 4 molecules-28-02620-f004:**
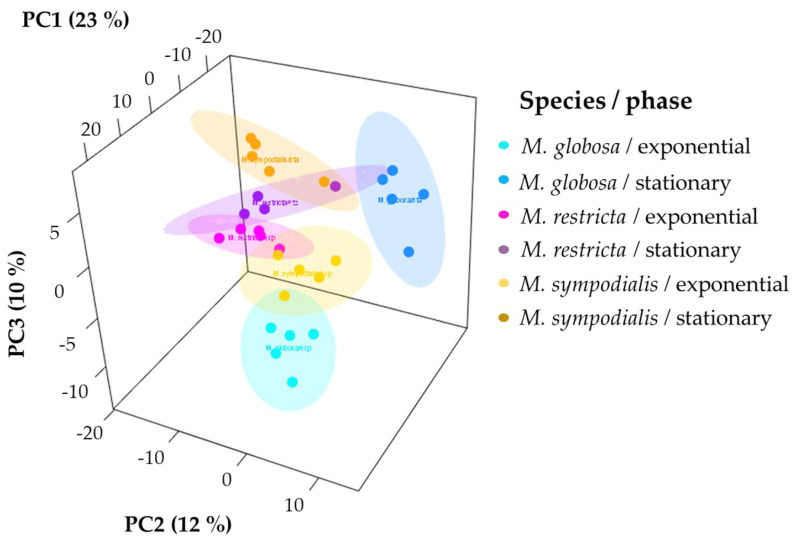
Principal component analysis (PCA) of the volatile profiles of *Malassezia* species in the exponential and stationary growth phases. The ellipses in this plot are confidence intervals of 95% using a normal distribution.

**Figure 5 molecules-28-02620-f005:**
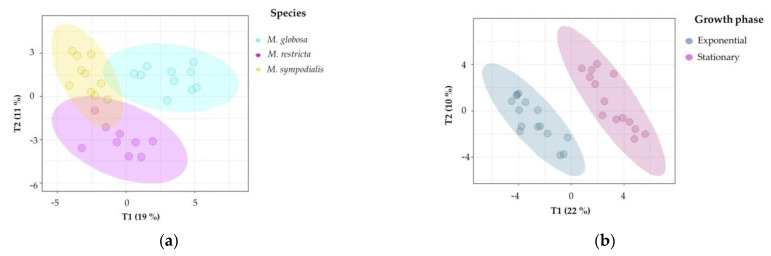
Projection of latent structures-discriminant analyses (PLS-DA) of the volatile profile for *Malassezia* species in two growth phases. (**a**) PLS-DA of the volatile profile discriminated by the species (Q2 = 0.798); (**b**) PLS-DA of the volatile profile discriminated by growth phases (Q2 = 0.877); (**c**) PLS-DA of the volatile profile discriminated by the species/growth phase (Q2 = 0.727); (**d**) Variable importance in projection (VIP) plot of the volatile profile discriminated by the species/growth phase. The ellipses in these plots are confidence intervals of 95% using a *t*-distribution.

**Figure 6 molecules-28-02620-f006:**
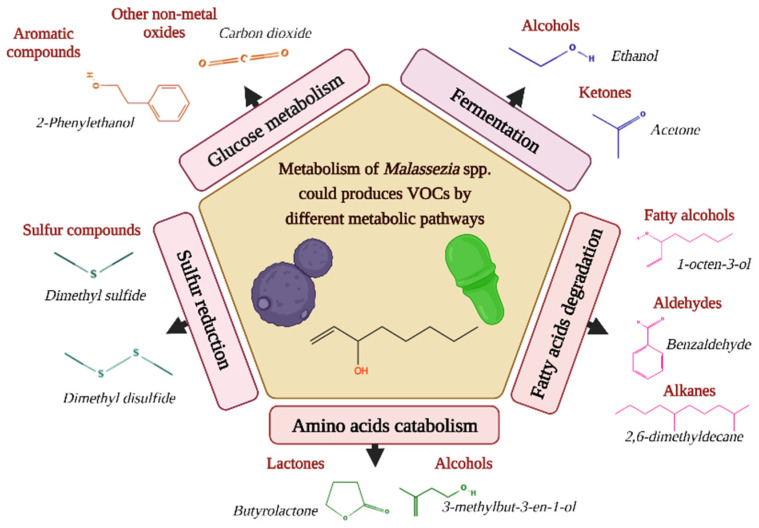
Different metabolic pathways can produce volatile organic compounds (VOCs) in *Malassezia* species. Representative examples are given per class of volatiles [13].

**Table 1 molecules-28-02620-t001:** Times of the exponential and stationary phases of *Malassezia* according to the growth curve.

	*M. globosa*	*M. restricta*	*M. sympodialis*
Exponential phase	30 h	30 h	10 h
Stationary phase	90 h	55 h	35 h

This table summarizes the specific times (presented in hours (h)) at which each phase occurs.

**Table 2 molecules-28-02620-t002:** Volatile profiles of the three species of *Malassezia* in the exponential and stationary growth phases.

No.	Compound	CAS	RI	RI Exp.	Ret. Time	ClassyFire Class/Subclass	*M. globosa*						*M. restricta*						*M. sympodialis*					
							Growth Phase																	
							Exp.			Stat.			Exp.			Stat.			Exp.			Stat.		
							% Area	SD	N	% Area	SD	N	% Area	SD	N	% Area	SD	N	% Area	SD	N	% Area	SD	N
1	Carbon dioxide	124-38-9			1.33	Other nonmetal oxides	13.89	5.7	5	11.17	4.2	5	11.28	1.3	5	22.78	5.6	4	7.06	2.1	5	23.67	2	5
2	Unknown 1				1.80		2.51	2.6	3	0.77		1	1.57		1							4.52	4.3	3
3	Propan-1-ol	71-23-8	568		1.82	Alcohols and polyols				3.81	2.8	4	3.09	3.1	3									
4	2-Methylpropanal	78-84-2	532		1.82	Carbonyl compounds										1.29		1	2.84	3.5	3			
5	Butane-2,3-dione	431-03-8	591		1.96	Carbonyl compounds							4.70	2.9	4				2.57	2.5	3			
6	Hex-1-ene	592-41-6	601		1.99	Unsaturated aliphatic hydrocarbons	0.88		1	1.96	1	5	0.92		1	0.53		1	1.60	2.2	2	0.50		1
7	2-Methylpropan-1-ol	78-83-1	614	626	2.26	Alcohols and polyols				4.35	3.7	4				8.73	5.5	4				7.80	3.9	5
8	Butan-1-ol	71-36-3	653	657	2.56	Alcohols and polyols	4.12	4.8	3	3.73	0.9	5	9.26	3.7	5	8.01	1.3	4				7.37	2.6	5
9	Pentan-2-one	107-87-9	695	684	2.80	Organooxygen compounds	3.24	2.7	5	1.45	0.6	5	3.12	2.5	5	5.45	1	4	1.09	1	3	1.89	0.4	5
10	Methylcyclohexane	108-87-2	755	720	3.28	Cycloalkanes	1.33	1.9	2	0.66	1	2	0.75		1				2.47	2.3	3	0.59		1
11	2,5-Dimethylhexane	592-13-2		726	3.38	Alkanes	0.29		1	0.12		1							0.46		1			
12	3-Methylbut-3-en-1-ol	763-32-6	731	727	3.40	Alcohols and polyols				0.23		1				1.03	0.8	3						
13	3-Methylbutan-1-ol	123-51-3	735	730	3.45	Alcohols and polyols	14.32	4.4	5	21.80	6.3	5	9.52	1.9	5	21.07	9.2	4	9.25	2.6	5	11.74	4.1	5
14	2-Methylbutan-1-ol	137-32-6	767	734	3.50	Alcohols and polyols				3.84	3.5	3	4.69	0.7	5	6.38	1.9	4	3.52	2	4	3.00	2.2	4
15	3-Methylpentan-2-one	565-61-7	759	748	3.74	Carbonyl compounds				2.74	0.7	5				0.98		1						
16	Pentan-1-ol	71-41-0	751	763	3.98	Alcohols and polyols				3.53	2.2	4	3.05	4.4	2	1.05		1				7.77	4.4	4
17	Unknown 2			768	4.06								0.46		1	0.54		1						
18	(E)-oct-3-ene	14919-01-8		797	4.54	Branched unsaturated hydrocarbons													0.31	0.4	2	0.11		1
19	(Z)-oct-3-ene	14850-22-7		805	4.72	Unsaturated aliphatic hydrocarbons	3.47	2.4	4	0.88	0.8	4	2.48	1.5	4	0.26	0.4	2	5.17	2.9	4	1.95	1.3	4
20	Unknown 3			813	4.90		3.93	2.7	4	1.32	1.4	3	3.64	1.6	5	0.83	0.8	3	8.28	1.1	5	2.34	1.9	4
21	2,4-Dimethylheptane	2213-23-2	818	819	5.08	Alkanes	15.00	8.7	4	1.44		1	4.07	5.6	2				12.42	3.9	5	2.62	2.4	3
22	(3E)-octa-1,3-diene	1002-33-1	827	822	5.14	Olefins													6.78	3.8	4	2.16	2.1	3
23	Propylcyclopentane	2040-96-2		828	5.30	Cycloalkanes	1.28	1.1	3	1.02	1.2	3	2.05	1.5	5	0.47	0.4	3	4.38	1.8	5	1.49	1.6	4
24	2,4-Dimethylhept-1-ene	19549-87-2	842	837	5.53	Branched unsaturated hydrocarbons	3.63	3.3	5	0.45	0.4	3	0.67	0.6	5	0.12		1	1.73	1.1	5	0.60	0.5	4
25	3-Methylhexan-2-one	2550-21-2		841	5.62	Carbonyl compounds				1.14	0.4	5				0.39		1						
26	Ethylbenzene	100-41-4	860	857	6.02	Benzene and substituted derivatives	0.23		1	0.39	0.4	3	0.52	0.8	2	0.57	0.4	3	1.43	0.6	5	0.62	0.4	5
27	Hexan-1-ol	111-27-3	861	865	6.22	Fatty alcohols	19.96	5.9	5	4.96	1	5	22.16	6.2	5	13.52	2.3	4	16.16	1.5	5	11.51	1.5	5
28	3-Methylbutyl acetate	123-92-2	884	873	6.44	Carboxylic acid derivatives	0.25	0.3	2	0.20	0	5	0.18		1	0.43	0.5	3						
29	3-Methylhexan-2-ol	2313-65-7		877	6.52	Alcohols and polyols				0.15	0.1	4							0.20		1	0.15		1
30	2,5-Dimethylpyrazine	123-32-0	911	909	7.42	Pyrazines				0.07		1	1.26	0.9	4	0.38	0.5	2				0.15	0.2	2
31	3-Propan-2-ylcyclohexene	3983-08-2		944	8.66	Unsaturated aliphatic hydrocarbons	0.38	0.3	3	0.04		1												
32	2,3-Dimethylcyclopent-2-en-1-one	1121-05-7		947	8.75	Olefins							0.18	0.2	3	0.03		1						
33	Unknown 4 (alkane)			958	9.13	Alkanes	2.71	0.6	5	0.36	0.1	5	2.59	0.8	5	1.16	0.5	4	2.73	0.3	5	1.33	0.3	5
34	Hept-2-en-1-ol	33467-76-4		965	9.40	Fatty alcohols				0.04	0.1	2												
35	Octan-3-one	106-68-3	987	984	10.06	Carbonyl compounds	3.36	2.3	4	0.03		1												
36	Octan-3-ol	589-98-0	994	994	10.40	Fatty alcohols				2.99	0.9	5												
37	2,6-Dimethylnonane	17302-28-2	1030	987	10.97	Alkanes	0.50		1	0.05		1	0.75	0.8	3	0.22		1	1.07	0.7	4	0.47	0.5	3
38	Unknown 5 (alkane)			996	11.03	Alkanes	0.46	0.6	2	0.26	0.1	5	1.77	0.6	5	0.82	0.3	4	1.94	0.4	5	0.95	0.2	5
39	2-Ethylhexan-1-ol	104-76-7	1039	1028	11.78	Fatty alcohols				18.18	11	4												
40	Unknown 6 (alcohol)			1081	13.96								0.48	0.7	2				1.12	0.2	5	0.37	0.6	2
41	2,6-Dimethyldecane	13150-81-7	1119	1103	14.87	Alkanes	2.46	2.4	3	0.47	0.2	5	3.08	1.0	5	1.36	0.4	4	3.25	0.6	5	1.29	0.3	5
42	2-Phenylethanol	60-12-8	1115	1111	15.21	Benzene and substituted derivatives				4.93	1.3	5				0.14		1				0.10		1
43	Heptyl acetate	112-06-1	1091	1112	15.29	Carboxylic acid derivatives	0.68	0.3	5							0.62	0.3	4	0.19	0.3	2			
44	Non-3-en-1-ol	51494-28-1	1126	1147	16.84	Fatty alcohols				0.05	0.1	3				0.03		1				1.53	0.5	5
45	Unknown 7			1153	17.08																	0.39	0.2	5
46	Unknown 8			1157	17.26		0.43	0.3	4	0.10	0	5	0.68	0.3	5	0.18	0.2	2	0.58	0.2	5	0.20	0.2	3
47	Isomer of dimethyldecane			1164	17.57	Saturated hydrocarbons	0.36	0.2	5	0.07	0.1	5	0.46	0.4	4	0.19	0.2	3	0.70	0.3	5	0.36	0.1	5
48	Unknown 9			1165	17.64								0.21	0.3	2	0.09	0.1	2	0.10		1	0.07	0.1	2
49	Unknown 10 (alcohol)			1176	18.12		0.06		1	0.04	0.1	2												
50	Unknown 11			1181	18.35					0.11	0.1	4	0.10	0.1	2	0.17	0.1	3	0.19	0.2	3	0.20	0	5
51	2,8-Dimethylundecane	17301-25-6		1225	20.25	Alkanes	0.29	0.3	3	0.05	0.1	3	0.25	0.3	3	0.10	0.1	2	0.37	0.1	5	0.15	0.1	4
52	Unknown 12 (alcohol)			1350	25.74					0.04		1				0.04		1						
53	Unknown 13 (alkane)			1371	26.60											0.03		1	0.05		1	0.02		1
54	Unknown 14			1471	30.75								0.01		1	0.05		1						

Exp. = exponential growth phase; Stat. = stationary growth phase; % Area = Average percentage area; SD = standard deviation; N = the number of replicates in which the compound is present; RI = Retention index published in literature; RI Exp. = Retention index relative to n-alkanes (C8–C20) on a BP-5 column (30 m × 0.25 mm × 0.25 µm); Ret. Time: retention time (min). Compounds were organized according to the ClassyFire class/subclass, which uses only chemical structures and structural features to automatically assign all known chemical compounds automatically https://classyfire.wishartlab.com/ (accessed on 21 June 2022) [34].

**Table 3 molecules-28-02620-t003:** Permutational multivariate analysis of variance applied to variables to compare the variance in the volatile profiles of *Malassezia* spp.

Variable	R^2^	*p* Value
Species	0.20	<0.001
Phase	0.20	<0.001
Species/phase	0.56	<0.001

## Data Availability

The data presented in this study are available within the article or Appendix A.

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
