# Peer review of "Why Do These Yeasts Smell So Good? Volatile Organic Compounds (VOCs) Produced by Malassezia Species in the Exponential and Stationary Growth Phases"

_molecules, 2023, doi:10.3390/molecules28062620_

Round 1
Reviewer 1 Report
The number of references is too large, 74. It may be reduced by removing older and less relevant references.
In contrast, the following literature should be reviewed and cited in the Introduction.
Learning about microbial language: possible interactions mediated by microbial volatile organic compounds (VOCs) and relevance to understanding Malassezia spp. Metabolism
Metabolomics. 2021; 17(4): 39.doi: 10.1007/s11306-021-01786-3
Many ways, one microorganism: Several approaches to study Malassezia in interactions with model hosts. PLOS Pathogens | https://doi.org/10.1371/journal.ppat.1010784 September 8, 2022
Lipid Metabolic Versatility in Malassezia spp. Yeasts Studied through Metabolic Modeling
Front. Microbiol. 8:1772. doi: 10.3389/fmicb.2017.01772
How does the yeast used in this study differs from the yeast available commercially? Do they produce similar volatile organic compounds (VOCs)?
One minor correction requires in Line 487, (0h, 12h, 24h, 27h, 30h, 33h, 36h, 48h, 60h, 72h, 96h, and 168h) should written as (0, 12, 24, 27, 30, 33, 36, 4h, 60, 72, 96, and 168 h)
Author Response
Dear reviewer,
We appreciate your comments and suggestions; we are sure these will improve our manuscript. Here we show a list of the corrections and answers to your question. Also, you can find the modifications in the manuscript highlighted in yellow.
-English language and style are fine/minor spell check required
To solve this topic, we sent the manuscript to an English-speaking academic expert for English review.
-The number of references is too large, 74. It may be reduced by removing older and less relevant references.
We understand your point. We consider suggestions pertinent, so we have changed or eliminated some old references about the topic studied in this research work. You can note the changes in the references list.
- In contrast, the following literature should be reviewed and cited in the Introduction.
Learning about microbial language: possible interactions mediated by microbial volatile organic compounds (VOCs) and relevance to understanding Malassezia spp. Metabolism
Metabolomics. 2021; 17(4): 39.doi: 10.1007/s11306-021-01786-3
Many ways, one microorganism: Several approaches to study Malassezia in interactions with model hosts. PLOS Pathogens | https://doi.org/10.1371/journal.ppat.1010784 September 8, 2022
Lipid Metabolic Versatility in Malassezia spp. Yeasts Studied through Metabolic Modeling
Front. Microbiol. 8:1772. doi: 10.3389/fmicb.2017.01772
According to your suggestion, we changed the old literature and included recent information about Malassezia research, so we reviewed the studies recommended. You can note these adjustments in the introduction of the document.
How does the yeast used in this study differs from the yeast available commercially? Do they produce similar volatile organic compounds (VOCs)?
The yeasts used in this study are reference strains that are also usually commercially available. To answer whether other Malassezia strains have a volatile profile similar to those studied here is challenging because there are no reports of volatile compounds for Malassezia species.
These data are available for M. furfur, whose volatile were compared to the species in this work, and some VOCs are similar.
So, we hope to set a precedent and open the gate to continue research on this topic in other Malassezia species or even in different isolates of the same species.
One minor correction requires in Line 487, (0h, 12h, 24h, 27h, 30h, 33h, 36h, 48h, 60h, 72h, 96h, and 168h) should written as (0, 12, 24, 27, 30, 33, 36, 4h, 60, 72, 96, and 168 h)
The correction was done
Reviewer 2 Report
Dear authors,
in my opinion, this work is definitely of interest to the reader of the Molecules Journal.
Author Response
Dear reviewer,
We appreciate that this research has aroused scientific interest and you consider it relevant for the journal.
We realized that you suggested that the references used and the presentation of the results could be improved, so we made some adjustments to these. We hope that with these modifications, the manuscript is clearer.
Reviewer 3 Report
Entitled: “Why do these yeasts smell so good? Volatile Organic Compounds (VOCs) produced by Malassezia species in the exponential and stationary growth phases”. The work determined the volatile profile of three Malassezia species in the exponential and stationary growth phases. The paper can be considered for publication. However, some suggestions should be considered and the revision was necessary.
1.The keywords and their order should be reconsidered carefully. The keywords are specific to the article, yet reasonably common within the subject discipline.
2.In the discussion section, an in-depth comparison and analysis of important results are needed. The discussion of the role of volatiles in biological interactions should be added. Therefore, we suggest that the authors carefully review this part and make it more focused and logical.
3.The cumulative percent of the three principal components only reach 45%, which cannot reach an acceptable level.
4.The English should be revised as there were certain inadequacies of the language expressions in the texts.
5.Abbreviations should be defined the first time they appear in each of three sections: the abstract; the main text; the first figure or table. Please double-check it.
6.Figure 1, The x-coordinate was in h instead of hours.
7.Table 2, The decimal point was represented by a dot instead of a comma.
Author Response
Dear reviewer,
We appreciate your comments and suggestions; we are sure these will improve our manuscript. Here we show a list of the corrections and answers to your question. Also, you can find the modifications in the manuscript highlighted in yellow.
- The keywords and their order should be reconsidered carefully. The keywords are specific to the article, yet reasonably common within the subject discipline.
We made some modifications in the order of these keywords, and some were changed for others we consider more appropriate.
- In the discussion section, an in-depth comparison and analysis of important results are needed. The discussion of the role of volatiles in biological interactions should be added. Therefore, we suggest that the authors carefully review this part and make it more focused and logical.
Concerning this suggestion, we considered that this is covered in the discussion. The first part of this issue is dedicated to comparing the results and explaining them from several views. Even we try to relate these findings to metabolism, phylogeny, and development. Moreover, the role of volatiles in interactions is discussed based on the conclusions of other research with other microorganisms, so for Malassezia, there is no report about the role of these compounds. We try to understand and correlate the production of some compounds with the biological function reported in other studies. However, further studies are necessary to elucidate the specific function of the compounds in interactions in Malassezia, so this is the first report of a volatile profile for M. globosa, M. restricta, and M. sympodialis.
-The cumulative percent of the three principal components only reach 45%, which cannot reach an acceptable level.
Although the analysis of principal components in other fields accepts higher percentages of variance, in metabolomics studies, it is common to find these analyses with similar percentages of variance due to the variation that the relative amounts of the metabolites found may have. Furthermore, more metabolites than grouping variables were obtained in the final data, which produced many dimensions in the PCA that explain the total variance. We took the first three dimensions to show how the species are grouped according to the volatile profile.
We can realize that other studies related to volatile organic compounds where performed the similar statistical analysis, and the value of variance are similar:
Comparison of Volatile and Nonvolatile Compounds in Rice Fermented by Different Lactic Acid Bacteria: doi: 10.3390/molecules24061183
- The English should be revised as there were certain inadequacies of the language expressions in the texts.
We checked the manuscript with an English-speaking academic expert for English review to solve this concern.
- Abbreviations should be defined the first time they appear in each of three sections: the abstract; the main text; the first figure or table. Please double-check it.
The review of abbreviations was performed and adjusted in the complete document.
- Figure 1, The x-coordinate was in h instead of hours.
The correction in figure 1 was done.
- Table 2, The decimal point was represented by a dot instead of a comma.
The correction in table 2 was done.